# Exploring the views and experiences of frailty and resilience among people experiencing homelessness in Ireland: A qualitative study

Thomas Cronin[1,2]*, Susan M. Smith[1], John Travers[1]

1 Discipline of Public Health and Primary Care, School of Medicine, Trinity College Dublin, Dublin, Ireland,
2 Irish College of General Practitioners, Ireland

* cronint2@tcd.ie

## Abstract

### Background

People experiencing homelessness (PEH) have high levels of frailty. However, their views on frailty, resilience, and the broader contextual factors that shape these experiences remain underexplored.

### Method

This was a qualitative study involving semi-structured interviews with 25 participants recruited from a primary care-based feasibility trial. Participants in the trial were adults aged 18 years and older with pre-frailty and frailty who were homeless. A convenience sampling strategy was used, with only participants who completed in-person follow-up invited to participate. Interviews were analysed thematically using Braun and Clarke's framework. The COREQ checklist guided study reporting.

### Results

Three main themes were identified from the data: (1) Frailty and homelessness – how homelessness both contributes to and magnifies frailty; (2) Survival – building resilience of PEH through coping mechanisms and informal support; and (3) Systemic neglect – the structural barriers and inadequate supports encountered by PEH.

### Conclusions

Frailty was experienced by participants as a multidimensional challenge, shaped by the realities of homelessness. Mental health was frequently described as central to the experience of frailty and should be prioritised in service responses. Addressing frailty in this population requires holistic approaches that consider structural disadvantage and the psychosocial context, alongside physical interventions.

**Data availability statement:** The data underlying this study consist of qualitative interview transcripts from people experiencing homelessness. Due to the vulnerable nature of the participant group, the small sample size, and the detailed narrative content of the interviews, there is a high risk of participant re-identification even after anonymisation. Public sharing of the full interview transcripts would therefore compromise participant confidentiality and ethical obligations. Data are available subject to approval from the Chair of the Irish College of General Practitioners Research Ethics Committee (contact via research@icgp. ie), for researchers who meet the criteria for access to confidential data.

**Funding:** ICGP Aspire Fellowship ICGP Research and Innovation grant We confirm that the funding sources (ICGP Aspire Fellowship; ICGP Research and Innovation Grant) had no role in study design, data collection and analysis, decision to publish, or preparation of the manuscript.

**Competing interests:** The authors have declared that no competing interests exist.

## 1. Introduction

Numbers of people experiencing homelessness (PEH) are increasing in Europe [1]. In Ireland, the compound annual growth rate of homelessness over the past decade is 13.5%, with numbers surpassing 16,000 people [2]. PEH have worse health outcomes and lower life expectancy compared to the general population [3,4]. Internationally, PEH have mortality rates that are twelve times higher than the general population for women and eight times higher for men [5]. Prior research has highlighted many of these premature deaths are from preventable and treatable conditions [6].

Qualitative research into the health of PEH has highlighted that this population frequently prioritise provision of basic human needs such as finding shelter and food, over seeking health and social care [7]. In addition, negative experiences around accessing healthcare services have been frequently reported [8–11]. This comes despite research indicating increased usage of acute healthcare services and emergency departments, as well as longer hospital stays amongst PEH [12,13]. Reasons for this are complex and multi-faceted with factors including psychological trauma, substance misuse and social exclusion postulated [6,10,14].

PEH are identified as encountering high levels of geriatric syndromes at an earlier age of onset compared to the general population, which includes frailty [15,16]. PEH experience frailty at a younger age and with higher prevalence than the general population [17,18]. This phenomenon has been described as 'premature' or 'accelerated' ageing [19], whereby biological age exceeds chronological age. This is likely the result of intersecting social and structural disadvantages, alongside homelessness itself, which contribute to sustained exposure to stress, illness, and barriers to timely care, which may cumulatively accelerate health decline in this population. This further heightens vulnerability in this group and diminishes their capacity to respond to health challenges, leading to an earlier onset of frailty.

Frailty is typically defined as a state of physiological vulnerability to external stressors, and is associated with elevated risks of illness, falls, functional dependency, disability, and mortality [20]. There are multiple measures of frailty [21], most of which are grounded in two dominant models. The phenotype model, established by Fried et al., defines frailty as the presence of three or more of the following criteria: exhaustion, weight loss, reduced muscular strength, slowed gait speed, and low physical activity [22]. In contrast, the cumulative deficit model, developed by Rockwood et al., conceptualises frailty as the accumulation of clinical signs, symptoms, diseases and disabilities, where the more deficits present, the greater the level of frailty [23]. While these models were developed primarily in older housed populations, their application to PEH raises important questions about how social and structural determinants of health are captured within existing frailty constructs.

In recent years, attention has also turned to resilience, often viewed as the conceptual counterpoint to frailty. Resilience reflects an individual's ability to withstand or recover from physiological stressors [24,25], and has been defined as "*the ability to recover or optimise function in the face of age-related losses or disease*" [26]. It is increasingly being recognised as a key element of successful ageing [27].

Existing qualitative literature on frailty has largely focused on the perspectives of older adults [28–30] and healthcare professionals [31–33]. Although PEH are known to experience high levels of frailty [17], research exploring their understanding of frailty and resilience remains limited. This represents a missed opportunity to embed the voices of PEH in the design and delivery of health and social care services aimed at addressing frailty [34], particularly given the distinct elements through which frailty may have developed and been experienced in this population.

As an initial step in exploring how frailty and resilience might be addressed in this population, a feasibility trial was conducted to test the acceptability and delivery of a brief intervention within a primary care setting for PEH [35]. As part of the trial's process evaluation, qualitative interviews were conducted with participants to explore their experiences and the broader context in which the intervention was delivered.

This paper presents a secondary analysis of those interviews, focusing specifically on participants' views and experience towards frailty and resilience. While the main outcomes of the trial are reported separately [35], this analysis provides an opportunity to foreground the lived experiences and insights of PEH in relation to these concepts. In doing so, it situates frailty not solely as a biomedical construct, but as one shaped by social, structural, and biographical forces relevant to public health practice and policy.

## 2. Methods

This study is a secondary analysis of qualitative data collected as part of the process evaluation for a feasibility trial involving people experiencing homelessness [35]. The trial tested a two-month intervention that included guided exercises, dietary information, and an oral nutritional supplement (ONS). The process evaluation followed a mixed-methods design, aimed at exploring participant engagement with the intervention and identifying contextual factors that influenced its delivery.

This paper draws specifically on the qualitative component – post-intervention semi-structured interviews – to examine participants' views on frailty and resilience and their beliefs on how frailty might be best addressed.

The secondary analysis was guided by a phenomenological informed approach, focusing on participants' lived experiences and personal meanings attached to frailty and resilience. The COnsolidated criteria for REporting Qualitative research (COREQ) checklist (S1 Table) was used to guide the reporting of this study [36]. Ethical approval was granted by the Irish College of General Practitioners (application number: ICGP_REC_2024_2082).

### 2.1. Recruitment and sampling

The study was conducted in an Irish primary care clinic for PEH. Over a five-month period between 13th August 2025 until 17th January 2025. All adults presenting to the clinic were screened for eligibility for the feasibility trial by two of the study authors (TC and JT), both general practitioners working at the site.

Inclusion criteria were:

- Aged ≥18 years at baseline

- Clinical Frailty Scale (CFS) score between 3 and 6 (pre-frail to moderately frail) [37]

- Current living circumstances classified under the "roofless," "houseless," or "insecure" categories according to the European Typology of Homelessness and Housing Exclusion (ETHOS) classification system for homelessness and housing exclusion [38]

- Capacity and willingness to provide informed consent

    Exclusion criteria included:

- Need for emergency medical care

- CFS < 3 or > 6

• Concurrent malignancy

• Chronic kidney disease (CKD) stage 3 or 4

• Coded diagnosis of severe dementia

• Inability to provide written informed consent

Participants were informed that they would be invited to a post-intervention interview as part of a process evaluation, and consent for this was obtained at the outset. Those who were followed up in person after completing the trial were eligible for interview and invited to participate, representing a convenience sampling approach.

## 2.2. Data collection

Data were collected through one-to-one semi-structured interviews, conducted in a private space at the clinic by a single interviewer (TC), who is a GP and the trial's principal investigator. TC had an existing clinical relationship with participants, which was acknowledged during the interview process and addressed through reflexive practice (see S2 Table).

Interviews were conducted face-to-face and audio-recorded with consent. Field notes were taken where appropriate. At the end of each interview, participants were asked to confirm whether they consented to their data being used in this secondary analysis. Interviews typically lasted around 10 minutes. This shorter duration reflected the constraints of the clinical environment and needs of the patients, where participants were attending for healthcare support and time was limited. Repeat interviews were not conducted.

The interview topic guide (S3 Table) included questions related to participant experiences of the intervention, perceptions of frailty and resilience, the impact of homelessness on frailty, and views on how frailty might best be managed. The guide was not formally pilot-tested but was reviewed and refined by the research team to ensure clarity and relevance. Semi-structured interviews were chosen to allow flexibility in phrasing and sequencing of questions, facilitating deeper engagement with individual participants [39].

All interviews were transcribed verbatim by the principal investigator. Transcripts were pseudo-anonymised using unique study IDs and stored securely on password-protected institutional servers. Transcripts were not returned to participants for review, as this was considered impractical given the study context.

## 2.3. Data analysis

Participant characteristics, including age bracket, gender, baseline Clinical Frailty Scale (CFS) score, opioid agonist treatment (OAT) status, recent illicit drug use (within the past six months), and ETHOS homelessness category, were recorded.

Interview transcripts were analysed using thematic analysis, following Braun and Clarke's six-step framework [40]. NVivo software supported the organisation and coding of the data. Codes were developed by TC and grouped into related categories. This was reviewed and refined through regular research meetings with co-author JT. Categories were refined into three overarching themes and associated sub-themes. This coding structure was developed iteratively during analysis to reflect the recurring patterns and relationships in the data. Themes were iteratively applied to subsequent transcripts and modified as needed.

A process of constant comparison, reflection, and engagement with raw data was used to ensure that the themes accurately captured participant views. As this was a secondary analysis of a process evaluation, the interviews were not originally designed to reach thematic saturation relating to participants' views toward frailty and resilience. However, themes related to these concepts became consistently evident across the dataset. As such, although the sample was not expanded until saturation was achieved, no new frailty- or resilience-related codes were observed to emerge in the later interviews, suggesting adequate coverage of the topic within this secondary analysis. Participants did not provide feedback on findings, as this was not feasible due to the secondary nature of the analysis.

## 3. Results

A total of 25 participants out of 108 (23.1%) recruited for the feasibility study participated in the qualitative process evaluation. Only those followed up in person (n = 54) were eligible to take part in the qualitative interviews, as telephone follow-up (n = 21) precluded interview participation. Of those followed up in person, 29 declined to participate in the recorded interview component, most commonly citing time pressures.

The included participants reflected heterogeneity across age, gender, frailty level, substance use, and housing circumstances (Table 1), providing a diverse cohort from which to explore participants' views. Participants ranged in age (28–60) and included both pre-frail and frail individuals, with varied experiences of homelessness spanning hostel accommodation, rough sleeping, and insecure housing. High levels of substance use and prescription of OAT were observed, reflecting the complex health and social profiles of the cohort.

While participants varied in age and sex, the themes identified were broadly shared across the sample, suggesting that views of frailty and resilience were shaped more strongly by homelessness and structural adversity than other factors such as age or gender. Three main themes with associated sub-themes emerged from the data (Fig 1), reflecting participants' views toward frailty and resilience: (1) Frailty and homelessness; (2) Survival; and (3) Systemic neglect. Illustrative quotations are included to support key findings.

### 3.1. Frailty and homelessness

The experience of homelessness, as expressed by participants, was seen to both contribute towards and magnify the impact of frailty. In the interviews, participants outlined how their physical and mental health deteriorated from living without stable housing and resulted in challenges for them in achieving better health and well-being. This illustrated the ongoing cycle of vulnerability that interviewees experienced, whilst also highlighting both the visible and invisible struggles they face due to their homeless status.

**3.1.1. Vulnerability.** Several participants highlighted the vulnerability that arose from living on the streets, including the need for constant vigilance due to the risk of violence, and how this worsened frailty:

*"I certainly had a number of incidents of injuries from violence that have increased my frailty... you never really fully get to rest. You're always in fight or flight mode. You're always walking the streets and you always kind of have to keep your guard up."* (**Participant 12**)

**Table 1.  Baseline characteristics of interview participants (n = 25).**

| Characteristic | n (%) or Mean (SD) [Range] |
|---|---|
| Age (years) | 43.9 (8.8) [28-60] |
| Sex | |
| Female | 6 (24.0) |
| Male | 19 (76.0) |
| Frailty status (CFS) | |
| Pre-frail | 15 (60.0) |
| Frail | 10 (40.0) |
| Accommodation status | |
| Hostel | 19 (76.0) |
| Rough sleeping | 3 (12.0) |
| Insecure accommodation | 3 (12.0) |
| Illicit substance use (past 6 months) | 18 (72.0) |
| Opioid agonist therapy prescribed | 18 (72.0) |

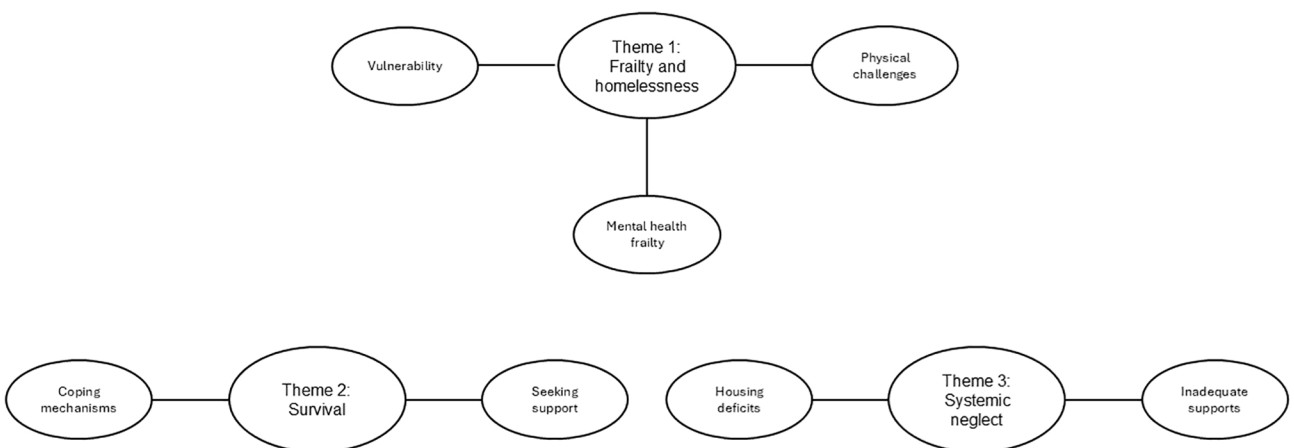

**Fig 1. Thematic map of identified themes and associated sub-themes.**

As well as difficulties with being able to rest in such a living situation, participants also revealed how the potential of physical violence impacted upon accessing adequate nutrition. This illustrated the barriers in place that can preclude strategies to overcome frailty, which include access to nutrition including sufficient protein.

> *"You don't like going to food banks or places where they give out food, because people on drugs, they're afraid, they get beaten up."* (**Participant 11**)

Furthermore, this vulnerability could create a sense of isolation and of becoming a target, which could further worsen frailty status and impact engagement with remedial services.

> *"Some people are living in tents, people get so used to being there on their own, they are loners, they keep away from someone and they like living in tents you know. But I've been there many times in my life, and you're vulnerable, you're weak, you're frail and people will take money away from you."* (**Participant 7**)

Alongside the vulnerability the arises from the threats of violence, vulnerability was also encountered from the harshness of homelessness. This included regular exposure to the elements and barriers to gaining a comfortable night's sleep. Multiple participants alluded to how living in this way contributed to accelerated ageing:

> *"If you're sleeping on the streets, if you've slept a year on the streets, it's been like you've gone about five years of your life, you know. It's a lot of energy. You're always listening, you know what I mean, you're always on the go. You get so tired sometimes, that you can just conk out."* (**Participant 3**)

> *"Like you just age faster. I'm like 50 something but I feel much older than I am. Like it's tough being on the streets, you can't sleep, you can't eat."* (**Participant 14**)

**3.1.2. Physical challenges.** The physical effect of participant's exposure to homelessness appeared to exacerbate and reinforce frailty through multiple pathways, including the negative health impact of substance misuse.

> *"Out on streets you're not eating, you're not looking after yourself. Like you're just thinking about your next fix you know. You can't sleep properly."* (**Participant 20**)

Malnutrition was described as a significant issue for participants, with weight loss and difficulties obtaining nutritious food frequently mentioned. One participant was able to describe a vivid picture of what frailty represented to him and his experience of it:

*"Frailty to me means... if I took my top off showed you, you'd know what frailty is! My hips are protruding. At my heaviest weight I was 14 stone and now I'm down to 9. My ribcage is showing. When I lie flat on the bed, like examination bed over there the doctors notice like my hips stick out... like I was showing signs of bed sores. I've been told I have a low body fat count. Like... I do eat... but I don't eat enough."* (**Participant 11**)

Barriers to cooking or accessing nutritious meals were also highlighted:

*"You can't eat properly. Like the places you go to for food... aren't good... most of the hostels you can't cook in."* (**Participant 15**)

Poor dentition also emerged as a potential contributor to frailty, with several participants discussing the pain and difficulties of eating, which could impact upon the types and quantity of food being consumed. Oral health frequently appeared to be neglected by participants as well as difficulties accessing appropriate dental care.

*"Like you don't want to eat... I have gum disease here. I was meant to get my teeth done in prison and all, have a deep clean and... I never got it done."* (**Participant 8**)

**3.1.3. Mental health frailty.** Several participants interconnected homelessness, frailty and mental health issues. The stress of homelessness often led to deteriorating mental well-being, which in turn made it harder to care for physical health. Participants often discussed the mental toll of being homeless and the associated stigma, which could create a barrier to seeking support or even engaging with others.

*"Your mental health goes down the drain. You're not really looking after yourself. You're frail in the mind more so than the body."* (**Participant 9**)

Others echoed this link between homelessness and declining mental health:

*"Like your mental health, you know, it goes really bad."* (**Participant 23**)

Some participants further highlighted how unresolved mental health issues could contribute to harmful coping strategies such as substance misuse:

*"A lot of people on the streets have mental issues like you know... and it doesn't really get sorted. And then you end up on drugs and it just kind of spirals."* (**Participant 11**)

The precarious living situation of participants was often discussed as having major impacts upon anxiety and feelings of hopelessness, whilst also worsening substance misuse. Mental health issues were also implicated in leading to homelessness.

*"I think it's... some people say it's all in your head, but I think it's like... sometimes it's dependent on how you become homeless. Like me, I become homeless because of my mental health. Maybe that's a bit of frail you know?"* (**Participant 5**)

## 3.2. Survival

Despite the overwhelming challenges of frailty and homelessness, participants frequently demonstrated resilience in their ability to adapt and survive. This was often a necessity in a response to their unstable living situation, whilst survival was also borne out of early life experiences such as childhood trauma. The support of others was seen as key to surviving, amongst interviewees, and psychological supports were frequently felt to be integral in combating frailty for PEH.

### 3.2.1. Coping mechanisms.
Resilience was generally seen as a positive concept amongst participants. Some participants viewed this as displaying mental perseverance, whilst for others the focus was more on the physical attributes that resilience brought about, such as being stronger and having increased levels of energy. One participant described how he was harnessing resilience to improve his physical condition.

*"Resilience means soldiering on. I'm trying to get back into the gym... you know they have benches and weights. I want to put muscle weight on. That's the plan. I'm only doing that now because of all the help I'm getting with addiction."* (**Participant 11**)

Other participants also highlighted the adversity in their childhood and demonstrated an ability to endure such a start in life and keep going. One participant alluded to the concept of intergenerational trauma and the potential cycle of homelessness.

*"What I went through with my step-dad, I got hidings, being thrown out on the street at 12. That probably impacted on how, what happened to my kids."* (**Participant 25**)

A reluctance to accept support, whether related to housing or drug treatment, was also discussed by participants and may reflect a form of self-protective coping. One participant described how seeking help around drug use could lead to social exclusion, highlighting the complex dynamics involved:

*"Like some people just don't want to be helped you know. Like some people just want to stay on the drugs and when you're trying to get clean you're out of the gang then and some people just don't wanna to do that."* (**Participant 20**)

### 3.2.2. Seeking support.
Interviewees highlighted the key role of support systems in fostering resilience. Several discussed the value of key workers, peer support groups and addiction clinics with helping them move forward. One participant reflected on the importance of having people to connect to.

*"You want to go into a place where you can walk in and there's a lot of people who want to help and... you know what I mean? And they'll work with you, you know what I mean and you can support them with going to meetings and going to key workers and things like that. When I was in … I had so many people's numbers I could ring up for support. Like if you relapse or having thoughts or things you know. All that counts, it does."* (**Participant 10**)

Another participant described how feeling supported by her keyworker made a significant difference in her ability to progress.

*"I think keyworkers can be good. Like I wanted to get help so it kind of made my keyworker want to help me more I think. And I think that helped me get out of the hostel I was in."* (**Participant 21**)

Participants also frequently emphasised the importance of psychological support in managing frailty in the context of homelessness. However, it was also pointed out that barriers to accessing these services, such as long waiting times and

limited availability, were often present. On being asking what supports might be helpful to people who are homeless and frail, one participant answered:

> "I think counselling should be much more easier to find. Sometimes you have to wait a long time to get counselling, things like that." (**Participant 5**)

### 3.3. Systemic neglect

A frequent sense of neglect was a recurrent theme in the accounts of interviewees. This included deficiencies in available emergency accommodation services, inadequate mental health supports and policies that many felt left them behind. These factors were identified as contributing to frailty and diminishing resilience and were highlighted as areas for improvement to mitigate the effects of frailty.

**3.3.1. Housing deficits.** The provision of adequate housing was frequently raised as a key issue to assist people who are homeless and frail. Some participants expressed frustration at housing shortages and perceived unfairness in the distribution of housing, capturing a sense of being left behind. In a couple of instances, this frustration extended to perceptions that housing was being prioritised for people seeking international protection, reflecting wider societal tensions around housing allocation.

> "They're giving out so many homes to people coming in, but there's still not enough. And they're doing their best, but the homeless are getting left behind, and it's getting bigger." (**Participant 10**)

Homeless hostels were also brought up by participants, and these were often a source of mixed experience. Some participants described instances of good support provided, but more frequently hostels were characterised as challenging and tough environments to live in. Indeed, a preference for staying in tents or living rough over hostels was expressed by some. One participant illustrated how hostels can represent barrier to recovery from addiction given the widespread use of drugs.

> "I'm in a hostel now… it's not a good hostel, drugs are going on in front me everyday. Everybody's doing it around me. So it's not good, and none of the staff in the hostel stop us, they don't give a shit." (**Participant 7**)

One participant reflected on the difficulty of trying to maintain recovery while surrounded by active drug use, describing a cycle of relapse linked to the lack of supportive environments:

> "Even when you get drug free, they send you back to the same place where you were, and you just get into drugs again." (**Participant 22**)

Furthermore, hostels were identified as being inappropriate for persons with high care needs, as might be the case for someone with frailty. One participant highlighted potential levels of dependency in a hostel and how staff were ill-equipped to deal with this.

> "Yeah, like proper people to look after you. Like some of the people I'm with should be in a care home. They are in such a condition that they can't look after themselves." (**Participant 22**)

**3.3.2. Inadequate services.** A lack of appropriate supports was frequently cited by participants as a barrier to overcoming frailty. These gaps ranged from the absence of tailored exercise programmes to the limited availability of structured mental health services:

> *"There should be more support for homeless people, more services, more psychologists, you know, especially more psychiatrists."* (**Participant 25**)

The inaccessibility of targeted services left many participants without tools to manage their health, compounding feelings of abandonment and despair. Harmful coping strategies such as substance misuse were described as responses to pain, boredom, and instability, often made worse by frequent interactions with law enforcement:

> *"I'm just looking for things that will kill the pain, just looking for things that will kill the time, so you end up on drugs. You just want to forget about everything. And you get a lot off the police you know, they never leave you alone."* (**Participant 8**)

Some participants also reflected on the limitations of existing medical treatments for drug addiction, such as long-term methadone use. While often essential for stabilisation, it was seen by some as physically depleting, highlighting the need for more holistic and sustained support drug services beyond pharmacological treatment alone.

> *"I've been on [methadone] for over 20 years… it just takes a lot away from you."* (**Participant 14**)

## 4. Discussion

### 4.1. Principal findings

This study is the first to explore the views and experiences of PEH who are pre-frail and frail towards frailty and resilience. Participants described homelessness as both contributing to and compounding frailty, through pathways such as exposure to violence, malnutrition, poor sleep, substance misuse, and limited access to healthcare. These accounts suggest that frailty among PEH may be influenced by cumulative life course exposures, including prolonged housing instability, adverse childhood events, repeated trauma, and disrupted access to institutional support, rather than chronological ageing alone.

Mental health was often central to participants' understanding of frailty, which could represent a consequence of homelessness and as well as a routes to homelessness. In this way, frailty was not framed purely as a physical state but one shaped by complex psychosocial challenges. Participants also expressed how difficult living environments, such as hostels or street settings, could accelerate physical and psychological decline. These narratives suggested possible biographical pathways into frailty, where early-life adversity, inadequately treated mental illness, and prolonged substance use may interact with homelessness to accelerate vulnerability and functional decline.

Despite these compounding risks, participants demonstrated notable resilience, often drawing on internal coping mechanisms and support from peers, keyworkers, and addiction services. Yet this resilience frequently occurred in spite of, rather than because of, systemic supports. Barriers to accessing appropriate services, including mental health care, secure housing, and tailored health interventions were frequently cited and contributed to feelings of neglect and reduced the capacity to overcome frailty.

### 4.2. Comparison to existing literature

Previous research has reported that older people often regard frailty negatively, viewed as an inevitable and irreversible state of decline [28,29,31,41], which may undermine engagement with interventions aimed at delaying or reversing frailty [42]. In our study, frailty was similarly described by participants in adverse terms, closely associated with weakness, vulnerability and accelerated ageing. Conversely, resilience was perceived positively, consistent with prior research describing it as a desirable attribute linked to physical strength and wellbeing [24,30].

Qualitative research with PEH has consistently identified barriers to healthcare access [8,10]. Our findings reinforce this, particularly in relation to mental healthcare. Participants frequently interlinked homelessness, frailty and mental

illness. This reflects wider evidence showing elevated frailty in those with serious mental illness [43,44], and the role of social isolation in the development of frailty [45].

Consistent with prior research [46,47] participants described emergency accommodation as frequently inappropriate, particularly for individuals with complex needs. Inadequate support, exposure to substance use, and lack of privacy or safety were identified as barriers to recovery, aligning with studies documenting how hostel environments can undermine addiction treatment efforts [48,49].

Oral health challenges were also prominently reported. This supports research showing that PEH experience disproportionately poor dental outcomes [50], often due to limited access to care [51], and the impact of substance use, such as methadone-induced xerostomia, on oral health [52]. Correspondingly, participants in this research reported issues with dentition and illustrated how this could be linked to frailty.

### 4.3. Strengths and limitations

All interviews were conducted by one interviewer which aided in consistency. Having prior knowledge, clinical experience and understanding of frailty helped inform the interviewer in interpreting and responding appropriately to participants' perspectives. However, the interviewer's role as a doctor was apparent to all interviewees and may have influenced the responses due to perceived power dynamics [53].

Participants were recruited from a feasibility trial of an intervention designed to reverse frailty and build resilience, which may have introduced selection bias in the sample. Notably, a number of individuals enrolled in the trial were lost to follow-up or declined to take part in the qualitative interviews. However, the final sample included a range of participants across age, gender, frailty status, and substance use backgrounds. As recruitment occurred within a healthcare-based setting, the sample is likely to over-represent individuals who are already engaged with healthcare services, which may limit generalisability to the wider homeless population. Nevertheless, recruitment was done during routine healthcare appointments, which means the sample reflects patients in this real-world setting.

Additionally, participants' involvement in the trial itself may have shaped their views on frailty and resilience, compared to what might have been expressed in a pre-intervention context. These factors limit the generalisability of the findings. Nonetheless, the study addresses a significant evidence gap regarding frailty in PEH, a population often underrepresented in research and provides new insight into the topic.

Lastly, transcripts were not returned to participants for comment or correction, and member checking was not conducted, representing a limitation in confirming participant interpretations. However, this decision was influenced by the transient nature of the study population, the secondary analysis of the data, and the constraints of the clinical environment in which data collection took place.

### 4.4. Implications for policy, practice and future research

This research demonstrates the structural barriers PEH face in attempting to overcome frailty. Living in homeless hostels or sleeping rough was frequently associated with disrupted sleep, contributing to a cycle of worsening mental health and substance misuse [48]. Moreover, such precarious living situations often prevent access to cooking facilities to produce nutritious protein-rich foods to delay or reverse frailty, whilst the risk of violence may impact upon home exercise programmes, both proven treatments in the management of frailty [54–56]. Interventions that fail to consider lack of access to institutional support, exposure to violence, mental illness, substance use and duration of homelessness in this group may have limited effectiveness. This illustrates the foundational importance of safe and secure housing to both protect health and enable engagement in evidence-based frailty interventions.

This study also identifies potentially under considered contextual factors for addressing frailty among PEH. Poor dentition and reduced access to cooking facilities, described by several participants, may significantly limit the ability to consume appropriate foods and should be considered when determining nutritional strategies. In certain cases, this

may justify the use of ONS, despite concerns about misuse in populations with substance use histories [57]. Additionally, improving access to dental care for this population should be prioritised in future health policy.

Given participants' strong emphasis on the interplay between mental health and frailty, improving access to psychological and psychiatric services is vital. In particular, there is an urgent need to develop dual diagnosis services, which address co-occurring mental illness and substance use frequently experience by PEH [58]. Existing reports note that "*there has been subdued response by policy-makers and services to providing effective health care to people with dual diagnosis in Ireland*" [59], and access remains limited and inconsistent across regions [60].

While dominant frailty models largely focus on physical deficits, our findings underscore the importance of social and psychological contributors to frailty. Future research should consider the development of frailty assessment tools tailored specifically to the context and experiences of PEH, which may better capture the domains most relevant to this group, enabling more targeted frailty treatments. The Tilburg Frailty Indicator [61], for example, incorporates physical, psychological, and social domains and may represent a more holistic framework that could be adapted or applied more widely in this population. Such approaches may offer more suitable means of detecting frailty compared to existing instruments primarily designed for older adults [17].

## 5. Conclusion

This qualitative study provides important insights into how PEH perceive frailty and resilience. Three key themes emerged: frailty and homelessness, survival, and systemic neglect. These illustrated how frailty is shaped and intensified by the lived experience of homelessness. Mental health was frequently identified as central to participants' understanding of frailty and should be a core element of any intervention targeting this group.

To address frailty effectively, interventions must extend beyond physical treatments to address structural factors such as housing instability, barriers to adequate nutrition, and limited access to appropriate mental health services, while acknowledging the adaptive, resilience strategies individuals develop in response to these challenges. Future research should prioritise the development of tailored frailty screening tools for PEH, incorporating psychological and social dimensions to guide more effective, targeted interventions.

## Supporting information

**S1 Table. Consolidated criteria for reporting qualitative studies (COREQ): 32-item checklist.** From: Tong A, Sainsbury P, Craig J. Consolidated criteria for reporting qualitative research (COREQ): A 32-item checklist for interviews and focus groups. Int J Qual Heal Care 2007.
(DOCX)

**S2 Table. Reflexivity statement.**
(DOCX)

**S3 Table. Interview topic guide.**
(DOCX)

## Acknowledgments

The authors would like to sincerely thank all participants for their time, engagement, and invaluable contributions to this study. We also extend our gratitude to all the staff at the Granby Clinic for their support and facilitation throughout the project.

## Author contributions

**Conceptualization:** Thomas Cronin, Susan M Smith, John Travers.

**Data curation:** Thomas Cronin.

**Formal analysis:** Thomas Cronin, Susan M Smith, John Travers.

**Investigation:** Thomas Cronin.

**Methodology:** Thomas Cronin, Susan M Smith, John Travers.

**Supervision:** Susan M Smith, John Travers.

**Writing – original draft:** Thomas Cronin.

**Writing – review & editing:** Susan M Smith, John Travers.

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
