## [Decision Letter · Decision Letter 0]

23 Jan 2026

Dear Dr. Cronin,

Thank you for submitting your manuscript to PLOS ONE. After careful consideration, we feel that it has merit but does not fully meet PLOS ONE’s publication criteria as it currently stands. Therefore, we invite you to submit a revised version of the manuscript that addresses the points raised during the review process.

We look forward to receiving your revised manuscript.

Kind regards,

Mario Ulises Pérez-Zepeda, M.D., Ph.D.

Academic Editor

PLOS One

**Journal Requirements:**

1. When submitting your revision, we need you to address these additional requirements. Please ensure that your manuscript meets PLOS ONE's style requirements, including those for file naming. The PLOS ONE style templates can be found at https://journals.plos.org/plosone/s/file?id=wjVg/PLOSOne_formatting_sample_main_body.pdf and https://journals.plos.org/plosone/s/file?id=ba62/PLOSOne_formatting_sample_title_authors_affiliations.pdf 2. Thank you for stating the following financial disclosure: ICGP Aspire FellowshipICGP Research and Innovation grant   Please state what role the funders took in the study.  If the funders had no role, please state: "The funders had no role in study design, data collection and analysis, decision to publish, or preparation of the manuscript." If this statement is not correct you must amend it as needed. Please include this amended Role of Funder statement in your cover letter; we will change the online submission form on your behalf. 3. Thank you for stating the following in the Acknowledgments Section of your manuscript: The authors would like to sincerely thank all participants for their time, engagement, and invaluable contributions to this study. We also extend our gratitude to all the staff at the Granby Clinic for their support and facilitation throughout the project. Finally, TC gratefully acknowledges the support of the Irish College of General Practitioners (ICGP) in the conduct of this research. We note that you have provided funding information that is not currently declared in your Funding Statement. However, funding information should not appear in the Acknowledgments section or other areas of your manuscript. We will only publish funding information present in the Funding Statement section of the online submission form. Please remove any funding-related text from the manuscript and let us know how you would like to update your Funding Statement. Currently, your Funding Statement reads as follows: ICGP Aspire FellowshipICGP Research and Innovation grant  Please include your amended statements within your cover letter; we will change the online submission form on your behalf. 4. Please upload a new copy of Figure 1, as the detail is not clear. Please follow the link for more information:  https://journals.plos.org/plosone/s/figures 5. We note that there is identifying data in the Supporting Information file. Due to the inclusion of these potentially identifying data, we have removed this file from your file inventory. Prior to sharing human research participant data, authors should consult with an ethics committee to ensure data are shared in accordance with participant consent and all applicable local laws. Data sharing should never compromise participant privacy. It is therefore not appropriate to publicly share personally identifiable data on human research participants. The following are examples of data that should not be shared: -Name, initials, physical address-Ages more specific than whole numbers-Internet protocol (IP) address-Specific dates (birth dates, death dates, examination dates, etc.)-Contact information such as phone number or email address-Location data-ID numbers that seem specific (long numbers, include initials, titled “Hospital ID”) rather than random (small numbers in numerical order) Data that are not directly identifying may also be inappropriate to share, as in combination they can become identifying. For example, data collected from a small group of participants, vulnerable populations, or private groups should not be shared if they involve indirect identifiers (such as sex, ethnicity, location, etc.) that may risk the identification of study participants. Additional guidance on preparing raw data for publication can be found in our Data Policy (https://journals.plos.org/plosone/s/data-availability#loc-human-research-participant-data-and-other-sensitive-data) and in the following article: http://www.bmj.com/content/340/bmj.c181.long. Please remove or anonymize all personal information (<specific identifying information in file to be removed>), ensure that the data shared are in accordance with participant consent, and re-upload a fully anonymized data set. Please note that spreadsheet columns with personal information must be removed and not hidden as all hidden columns will appear in the published file. 6. Please include captions for your Supporting Information files at the end of your manuscript, and update any in-text citations to match accordingly. Please see our Supporting Information guidelines for more information: http://journals.plos.org/plosone/s/supporting-information. 7. If the reviewer comments include a recommendation to cite specific previously published works, please review and evaluate these publications to determine whether they are relevant and should be cited. There is no requirement to cite these works unless the editor has indicated otherwise. 

Reviewers' comments:

**Comments to the Author**

1. Is the manuscript technically sound, and do the data support the conclusions?

Reviewer #1: Yes

Reviewer #2: Partly

2. Has the statistical analysis been performed appropriately and rigorously?

Reviewer #1: N/A

Reviewer #2: N/A

3. Have the authors made all data underlying the findings in their manuscript fully available?

Reviewer #1: No

Reviewer #2: No

4. Is the manuscript presented in an intelligible fashion and written in standard English?

Reviewer #1: Yes

Reviewer #2: Yes

**Reviewer #1:**  I recommend the publication of the paper as it fills a gap in the literature:

It explores how pre-frail and frail PEH themselves understand frailty and resilience. (Maybe a small weakness in the sample is that all participants were drawn from a feasibility trial and already engaged with healthcare, creating a potential bias toward individuals who are more health-engaged than the general homeless population).

It extends frailty research beyond older adults and clinical populations to a younger, socially marginalized group with high premature frailty.

It contributes to homelessness research by linking structural conditions directly to frailty trajectories, so shifting the focus from individual frailty traits toward structural determinants and lived context,.

It challenges dominant frailty constructs by demonstrating that psychological and social dimensions are central for PEH, implying that existing frailty tools may not be appropriate.

2. Statistical Analysis: It is a qualitative study, a convenience sampling strategy was used

3. Data availability: No. some restrictions will apply

4. Ethics: Ethical approval was granted by the Irish College of General Practitioners (application number: ICGP_REC_2024_2082).

**Reviewer #2:**  1.Stronger problematization of the context: While the introduction provides a solid overview of frailty and homelessness, it could be strengthened by a more explicit problematization of why it is particularly important to analyse frailty in this population. Expanding on the social, structural, and biographical mechanisms that lead to accelerated ageing and vulnerability among people experiencing homelessness would help situate the study more clearly within a critical public health perspective.

2.Descriptive characterization of the sample: I suggest adding an additional descriptive table showing participant characteristics by sex and age groups, both at baseline and follow-up. This would allow readers to better understand the heterogeneity of the sample and contextualize the narratives. Variables such as age, sex, frailty level, substance use, and housing situation could be included.

3.Deeper analytical use of the narratives: The results section is coherent and well organized, but the analysis could be strengthened by more systematically exploring similarities and divergences across participants, particularly: By age groups (younger vs. older participants). By sex. This comparative perspective could reveal meaningful patterns in how frailty and resilience are experienced and interpreted.

4.Linking narratives to life trajectories and explanatory factors:The discussion could benefit from a more interpretive reflection on the factors potentially associated with the observed narratives (e.g., life course experiences, duration of homelessness, access to institutional support, exposure to violence, mental health, substance use). This would enhance the theoretical contribution of the study and increase its relevance for intervention design and policy.

**Do you want your identity to be public for this peer review?** For information about this choice, including consent withdrawal, please see our Privacy Policy

Reviewer #1: No

Reviewer #2: No

---

## [Author Response · Author response to Decision Letter 1]

30 Jan 2026

Please see attached Word Document uploaded as 'Response to reviewers'.

---

## [Editor Report · Decision Letter 1]

5 Feb 2026

Exploring the views and experiences of frailty and resilience among people experiencing homelessness in Ireland: a qualitative study

PONE-D-25-58699R1

Dear Dr. Cronin,

We’re pleased to inform you that your manuscript has been judged scientifically suitable for publication and will be formally accepted for publication once it meets all outstanding technical requirements.

Kind regards,

Mario Ulises Pérez-Zepeda, M.D., Ph.D.

Academic Editor

PLOS One
---

## [Editor Report · Acceptance letter]

PONE-D-25-58699R1

PLOS One

Dear Dr. Cronin,

I'm pleased to inform you that your manuscript has been deemed suitable for publication in PLOS One. Congratulations! Your manuscript is now being handed over to our production team.

Kind regards,

on behalf of

Dr. Mario Ulises Pérez-Zepeda

Academic Editor

PLOS One